# Identifying patients with multidrug-resistant tuberculosis who may benefit from shorter durations of treatment

**Nicholas Winters**[1], **Mireille E. Schnitzer**[1,2,3], **Jonathon R. Campbell**[4,5,6], **Susannah Ripley**[1], **Carla Winston**[7], **Rada Savic**[8,9], **Nafees Ahmad**[10], **Gregory Bisson**[11], **Keertan Dheda**[12], **Ali Esmail**[12], **Medea Gegia**[13], **Ignacio Monedero**[14], **Margareth Pretti Dalcolmo**[15], **Denise Rodrigues**[16], **Rupak Singla**[17], **Jae-Joon Yim**[18], **Dick Menzies**[1,5]\*

1 Department of Epidemiology, Biostatistics and Occupational Health, McGill University, Canada, 2 Faculty of Pharmacy, Université de Montréal, Montreal, Canada, 3 Department of Social and Preventive Medicine, Université de Montréal, Montreal, Canada, 4 Department of Medicine & Department of Global and Public Health, Faculty of Medicine and Health Sciences, McGill University, Montreal, Canada, 5 McGill International TB Centre, Montreal Chest Institute, Research Institute of the McGill University Health Centre, Montreal, Canada, 6 Respiratory Epidemiology and Clinical Research Unit, Centre for Outcomes Research & Evaluation, Research Institute of the McGill University Health Centre, Montreal, Canada, 7 US Centers for Disease Control and Prevention, Atlanta, Georgia, United States of America, 8 Department of Bioengineering and Therapeutic Sciences, University of California San Francisco Schools of Pharmacy and Medicine, San Francisco, California, United States of America, 9 UCSF Center for Tuberculosis, University of California San Francisco, San Francisco, California, United States of America, 10 Faculty of Pharmacy and Health Sciences, University of Baluchistan, Quetta, Pakistan, 11 Department of Medicine and Department of Biostatistics, Epidemiology, and Informatics, Perelman School of Medicine at the University of Pennsylvania, Philadelphia, PA, United States of America, 12 Centre for Lung Infection and Immunity, Department of Medicine & UCT Lung Institute, University of Cape Town, Cape Town, South Africa, 13 Global Tuberculosis Program, World Health Organization, Geneva, Switzerland, 14 TB-HIV Department, International Union against Tuberculosis and Lung Diseases, Paris, France, 15 Centro de Referência Helio Fraga, Fiocruz, Brazil, 16 Instituto Clemente Ferreira, Sao Paulo, Brazil, 17 National Institute of Tuberculosis & Respiratory Diseases, New Delhi, India, 18 Department of Internal Medicine, Division of Pulmonary and Critical Care Medicine, Seoul National University College of Medicine, Seoul, South Korea

\* dick.menzies@mcgill.ca

**Data Availability Statement:** The data is accessible through University College London who now hold the data as a publicly accessible

## Abstract

### Objective

Studying treatment duration for rifampicin-resistant and multidrug-resistant tuberculosis (MDR/RR-TB) using observational data is methodologically challenging. We aim to present a hypothesis generating approach to identify factors associated with shorter duration of treatment.

### Study design and setting

We conducted an individual patient data meta-analysis among MDR/RR-TB patients restricted to only those with successful treatment outcomes. Using multivariable linear regression, we estimated associations and their 95% confidence intervals (CI) between the outcome of individual deviation in treatment duration (in months) from the mean duration of their treatment site and patient characteristics, drug resistance, and treatments used.

repository. Access to this is governed by an oversight committee and applications must be directed to that committee. There are legal restrictions in terms of signed data sharing agreements with all data contributors. Inquiries can be made here: https://www.ucl.ac.uk/global-health/research/tb-ipd-platform.

**Funding:** Data were contributed by the Tuberculosis Epidemiologic Studies Consortium of CDC and by other CDC projects. The findings and conclusions in this report are those of the authors and do not necessarily represent the official position of the Centers for Disease Control and Prevention.

**Competing interests:** The authors have declared that no competing interests exist.

## Results

Overall, 6702 patients with successful treatment outcomes from 84 treatment sites were included. We found that factors commonly associated with poor treatment outcomes were also associated with longer treatment durations, relative to the site mean duration. Use of bedaquiline was associated with a 0.51 (95% CI: 0.15, 0.87) month decrease in duration of treatment, which was consistent across subgroups, while MDR/RR-TB with fluoroquinolone resistance was associated with 0.78 (95% CI: 0.36, 1.21) months increase.

## Conclusion

We describe a method to assess associations between clinical factors and treatment duration in observational studies of MDR/RR-TB patients, that may help identify patients who can benefit from shorter treatment.

## Introduction

Multidrug-resistant tuberculosis (MDR-TB), defined as tuberculosis with resistance to both rifampicin and isoniazid, is a major global health burden [1]. Although treatment success has increased over time to 60–70% [1, 2], the estimated number of MDR-TB cases has increased from previous years to 450,000 in 2021 [1]. Current recommended treatment from the World Health Organization (WHO) for extensive or severe MDR-TB is as long as 18–20 months [3] and entails a high patient burden. There is no doubt that shorter regimens are attractive for patients, health systems, and providers, as they reduce the burden of treatment [4–7]. In the past 10 years, several studies [8–12] have investigated shorter regimens for treatment of MDR-TB in randomized controlled trials (RCTs), but these may not reflect treatment in programmatic settings.

Assessing the effect of MDR-TB treatment duration in non-randomized studies has several potential limitations. Individuals' treatment durations are determined by the outcomes of loss to follow-up, failure, and death. For those remaining on treatment, the regimens and duration are highly individualized and vary by provider and patient presentation, which entail methodological challenges. Despite these challenges, investigators have used individual duration as an outcome [13], but inferences were limited and the evidence is considered by the WHO to be of very low quality [14].

Based on previous analyses using individual patient data (IPD) [13, 15] treatment duration varies widely between treatment sites and each site typically has a 'usual' duration of treatment targeted for patients, which may be based on local guidelines, experience, patient population, and availability of anti-tuberculosis drugs. However, there is substantial individual variation around that usual duration at each site. We hypothesize that analyzing individual differences from the site-specific mean treatment duration, among patients with successful treatment outcomes, may help address these methodologic challenges.

Our aim was to describe associations with site-specific average treatment durations and to use deviations from these site-specific average treatment durations to identify clinical and treatment factors associated with shorter duration of treatment among individual rifampicin-resistant (RR-TB) and MDR-TB patients with successful treatment outcomes.

## Methods

We conducted this study using a dataset of the 2019 IPD in MDR/RR-TB described in detail previously [15]. This study began in 2016 when the dataset was initially assembled. Briefly, the dataset included data from studies conducted between January 1, 2009, and April 15, 2016 that were identified in a systematic review [16]. In addition, the IPD were updated with data contributed by authors of a 2010 IPD meta-analysis [17] and data from two public calls by the WHO in 2018 [18] and 2019 [19] (Supplemental Figure S1 in S1 File presents a timeline for important changes in WHO treatment guidelines for MDR-TB). For comprehensive details on search strategy, study eligibility, and quality assessment see Supplement 1 in S1 File. Studies exclusively in children were excluded.

The 2019 IPD in MDR/RR-TB contains records from 55 studies and 13,272 patients who initiated treatment between 1993 and 2019 in 38 countries and regions. The characteristics of studies included in the IPD have been described previously [20, 21] and the quality and completeness of all studies in the IPD are described in Supplement 2 (S1 File).

### Study population

We included studies reporting individual treatment duration and excluded studies which did not provide information on duration, or only provided planned durations. From the included studies, we included only patients that had successful (cured or completed) treatment outcomes, as defined elsewhere [22, 23] and who had their individual treatment duration recorded. We verified outcomes provided by study investigators in their original study, and harmonized these to WHO 2013 definitions [23], as detailed elsewhere (see supplement to Ahmad et al. [15]). In those with death, failure, or loss to follow-up, their treatment duration is determined by their outcome, which may bias associations between characteristics and treatment duration, and were thus excluded. Any patients for which their individual treatment duration was missing were excluded from our primary analyses.

### Outcomes

We assessed two outcomes among patients with successful treatment outcomes: i) the mean treatment duration at each treatment site, which was used in an ecological level analysis to explore potential associations with site-level factors; and ii) the difference between each individual's treatment duration and the mean treatment duration of all patients with treatment success at their site. The latter is our primary outcome in this analysis, which is the individual deviation from the site-specific mean treatment duration; this is referred to as *deviation in treatment duration* throughout the text and interpreted in terms of shorter (negative value) or longer (positive value) duration of treatment in months.

### Statistical analysis

**Ecological analysis of mean treatment duration of site.** We first conducted an ecological analysis of the site-specific mean treatment duration in patients with successful outcomes where the unit of analysis was the treatment site, rather than the individual patient. Using available (non-imputed) data, we computed site-level proportions of categorical variables and means of continuous variables and described all using mean and standard deviation (SD), median and interquartile range [IQR], and range (minimum to maximum). We then performed univariable and multivariable linear regression in imputed data (described below) to examine associations between site-level characteristics and the mean treatment duration of the site (see Supplement 3 in S1 File for details).

**Analysis of individual deviation from mean treatment duration of site.** In our primary analysis, our approach was to construct an exploratory, hypothesis generating, multivariable model to identify factors conditionally associated with a change in deviation in treatment duration, while controlling for all variables selected into the model.

For clinical characteristics, drug susceptibility testing results, and treatments used we described categorical variables as n (%) while continuous variables were described using mean and standard deviation (SD) or median and interquartile range (IQR) using the available data (for detail on all variable specifications see Supplement 4 in S1 File). We also presented the regression coefficients (in months) and their 95% CI for age- and sex-adjusted univariable associations between deviation in treatment duration and each variable listed previously.

All regression analyses were conducted using data imputed with multivariate imputation by chained equations (MICE) with the assumption that data were missing at random (see Supplement 5 in S1 File for detail). The deviation in treatment duration was imputed for those with either only planned or missing deviation in treatment duration for our sensitivity analyses, along with the other variables, however we only included subjects with non-missing duration in our primary analysis. Twenty data sets were generated with 25 Gibb's sampling iterations [24].

To construct our exploratory model, we included variables known to be associated with treatment success in the published literature [13, 15, 20, 21]. Additionally, we ran adaptive Lasso regression [25], using each imputed data set, on the previously listed characteristics to identify other potentially important predictors of treatment duration that were not a priori identified. Pearson coefficients were used to assess correlation between variables to be included. When highly correlated variables were present, we chose the more clinically relevant variable. We then used multivariable linear mixed-effects models with a random intercept for study to estimate regression coefficients and 95% confidence intervals (CI) for each selected covariate, controlling for the others.

In subgroup analyses, we assessed the final model stratified by subpopulations of patients: i) with MDR/RR-TB plus resistance to both fluoroquinolones (FQ) and second-line injectables (SLI) and all others with MDR/RR-TB (including resistance to FQ or SLI but not both); ii) with or without extensive disease (defined as yes if acid-fast bacilli (AFB) smear positive at baseline, and if AFB smear status was missing then the presence of radiographic findings of cavitation or bilateral disease); and iii) with or without previous tuberculosis treatment. We also did additional exploratory analyses in subgroups of those with: i) extensive disease with only MDR/RR-TB and those without extensive disease with MDR/RR-TB plus any additional resistance; and ii) those with past tuberculosis treatment with MDR/RR-TB only and those without any past treatment with MDR/RR-TB plus any additional resistance. Additionally, we explored the possible effect of selection bias on our population by analyzing our final model adjusted with inverse probability of selection weights for inclusion into the study population (see Supplement 6 in S1 File for detail). We also performed an analysis that included subjects with missing treatment durations whose durations were imputed in the MICE procedure. Finally, we explored the impact that unmeasured confounding may have on the largest associations estimated from our primary analysis by calculating E-values as described by VanderWeele et al. [26] (see Supplement 7 in S1 File). All analyses were conducted using R version 4.1.2. [27]

This study used individual patient data provided by the investigators of the original studies, who obtained informed consent from all participants as appropriate for their original study designs. All data received were anonymized. This analysis received ethical approval from the McGill University Health Centre Research Ethics Board. Ethics approval was also obtained at participating sites, if considered necessary.

## Results

Of the 13,272 patients from 55 studies in the entire IPD, we included 6,702 from 49 studies that included 84 treatment sites in 34 countries (Fig 1). We excluded 6,570 patients in total. Six entire studies were excluded (2,235 patients) as they provided only planned duration or did not provide duration data (excluded and included studies were similar, see Supplement 2 in S1 File). Of the included studies, 4,335 patients were excluded: 44 had success but no duration data and 4,291 did not have treatment success. The characteristics of patients excluded from our analysis are presented in Supplementary Table S1 in S1 File.

### Ecological analysis of mean treatment duration of site

Descriptions of the site-level characteristics are presented in Table 1. The mean treatment duration of all sites was 22.8 and ranged from 12 to 36 months (see Supplementary Table S2 in S1 File for mean treatment duration of each site).

In univariable analysis, the proportion of patients at the site with past first-line drug use, MDR/RR-TB plus resistance to both FQ and SLI (MDR-FQ+SLI), or resistance to pyrazinamide were associated with longer mean treatment duration at the site. However, in multivariable analysis, only the proportion of patients with MDR-FQ+SLI was associated with longer mean treatment duration of site (Table 1).

### Analysis of individual deviation from mean treatment duration of site

The patients included in this analysis are described in Tables 2 and 3. The average total treatment duration was 22.0 months with SD of 4.6 (median 22 [IQR: 19, 24]). In univariable analyses, lower body mass index, past first- and second-line drug use, cavitation or bilateral disease on X-ray, and AFB smear positivity were all associated with longer treatment duration. Resistance to each drug, if tested (except linezolid, which was rarely tested) was associated with longer treatment duration. Longer treatment duration was associated with MDR/RR-TB plus resistance to SLI but FQ sensitive (MDR-SLI), or MDR/RR-TB plus resistance to FQ but SLI sensitive (MDR-FQ), or MDR-FQ+SLI. Within the treatment regimen of a patient, the use of capreomycin, kanamycin, moxifloxacin, levofloxacin, PAS, linezolid, clofazimine, Amx-Clv, clarithromycin, or bedaquiline, as well as greater number of drugs, were all associated with longer treatment duration in univariable analyses.

In the final multivariable model (see Supplement 8 in S1 File for detail on variable selection due to correlation of variables), longer treatment duration was associated with presence of cavitation, AFB smear positivity, HIV infection, past first-line drug use, and MDR/RR-TB with all types of additional resistance (Fig 2). Individual deviation from mean duration of site was also associated with several treatment factors. In contrast to univariable regression results, use of bedaquiline was associated with **shorter** treatment duration by -0.51 (95% CI -0.87 to -0.15) months in adjusted analyses. Longer treatment duration was associated with use of clarithromycin (1.12 months; 95% CI 0.71, 1.53), and with greater number of drugs used, or use of moxifloxacin, kanamycin, capreomycin, or Amx-Clv.

Results were similar when using inverse probability weights for selection into our study population from the entire IPD (Supplementary Table S3 in S1 File). However, in our sensitivity analysis including patients whose treatment durations were imputed, results were substantially different (Supplemental Figure S2 in S1 File).

E-values for the largest regression coefficients from our primary analysis are presented in Supplementary Table S4 (S1 File). For bedaquiline, an unmeasured confounder would need to have a risk ratio associated with both use of bedaquiline and treatment duration of 1.50 to completely explain away the association we observed with bedaquiline. The largest E-value

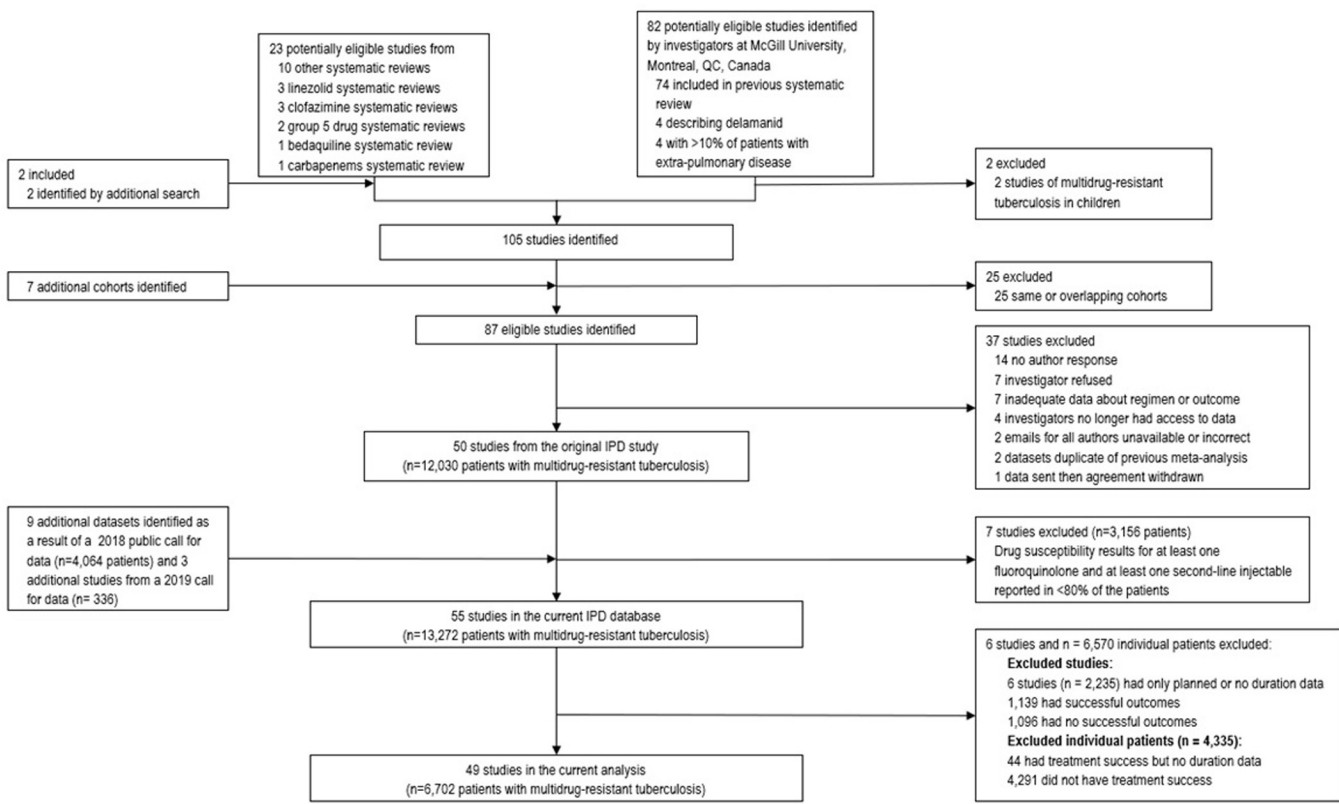

**Fig 1. PRISMA diagram for studies and patients included and excluded from the study population.**

required of an unmeasured confounder to explain away our estimated associations was for use of clarithromycin, while the smallest was for cavitation.

**Subgroup analyses.** In subgroup analyses (Table 4) the direction of associations between shorter treatment duration and use of bedaquiline remained consistent across all subgroups (except in those with MDR-FQ+SLI), and regardless of disease extent.

Associations between bedaquiline and duration were similar between those with or without past treatment. Additionally, use of Amx-Clv and clarithromycin were consistently associated with longer treatment duration in all subgroups. Body mass index was not associated with treatment duration in any subgroup while HIV was associated with longer duration in those with extensive disease and past tuberculosis treatment.

In other exploratory analyses (Supplemental Table S5 in S1 File) results were similar for bedaquiline, Amx-Clv, clarithromycin, and body mass index. However, HIV was not associated with treatment duration in any exploratory subgroup.

## Discussion

With this IPD meta-analysis of 6,702 MDR/RR-TB patients with treatment success, we have applied a novel approach to identify patients who may benefit from shorter MDR/RR-TB treatment. In ecological analysis of site-level factors, the only clinical or treatment characteristic associated with average treatment duration of a site was the proportion of MDR patients with added resistance to FQ and SLI. The lack of associations between mean treatment duration of site with many clinical factors (such as age, HIV infection, past treatment, or other patterns of drug resistance) may indicate that unmeasured factors like physician beliefs, site

**Table 1. Site-level characteristics and their univariable and multivariable associations with the site-specific mean treatment duration in patients with successful treatment outcomes.** Estimates and 95% confidence interval (CI) from linear regression models.

| Variable: proportion at site unless stated otherwise (n = 84) | Mean (SD) | Median [IQR] | Range | Mean site-specific treatment duration | |
|---|---|---|---|---|---|
| | | | | Univariable Months (95% CI) | Multivariable Months (95% CI) |
| **Clinical characteristics** | | | | | |
| Age (mean years) | 37.5 (6.1) | 37.6 [33.4, 41.5] | 21 to 55 | 0.01 (-0.1, 0.2) | 0.1 (-0.1, 0.3) |
| Sex (Female) | 0.37 (0.2) | 0.38 [0.27, 0.48] | 0 to 1 | -3.5 (-7.8, 0.9) | -2.7 (-7.9, 2.5) |
| HIV infection | 0.09 (0.2) | 0 [0, 0.09] | 0 to 0.73 | 3.0 (-1.9, 7.9) | 3.0 (-2.9,.8.8) |
| 2018 World Bank income category | | | | | |
| Low/lower-middle income—n (%) | 13 (15.5) | NE | NE | Ref | Ref |
| Upper-middle income—n (%) | 31 (36.9) | NE | NE | -1.4 (-4.1, 1.2) | -1.7 (-4.7, 1.2) |
| High income—n (%) | 40 (47.6) | NE | NE | -2.1 (-4.7, 0.4) | -1.5 (-4.9, 1.9) |
| Extensive disease* | 0.72 (0.24) | 0.74 [0.52, 0.95] | 0 to 1 | 1.4 (-2.3, 5.2) | 1.0 (-3.4, 5.4) |
| **Treatment history and drug resistance** | | | | | |
| Past first-line TB drugs | 0.70 (0.29) | 0.79 [0.47, 0.98] | 0 to 1 | 3.3 (0.2, 6.5) | 1.3 (-4.1, 6.7) |
| Past second-line TB drugs | 0.25 (0.32) | 0.08 [0, 0.5] | 0 to 1 | 2.3 (-0.6, 5.1) | -0.2 (-5.4, 4.9) |
| Number of effective drugs used | 4.41 (0.78) | 4.27 [4, 4.9] | 2.4 to 7.08 | -0.8 (-2.0, 0.3) | -0.2 (-1.6, 1.2) |
| MDR/RR-TB + FQ & SLI sensitive | 0.51 (0.36) | 0.41 [0.18, 0.9] | 0 to 1 | 3.0 (0.5, 5.5) | |
| MDR/RR-TB + FQ resistant & SLI sensitive | 0.14 (0.20) | 0.05 [0, 0.19] | 0 to 1 | 1.2 (-3.2, 5.6) | 0.1 (-5.2, 5.5) |
| MDR/RR-TB + SLI resistant & FQ sensitive | 0.13 (0.17) | 0.08 [0, 0.18] | 0 to 1 | -2.4 (-7.4, 2.7) | -0.2 (-6.3, 6.0) |
| MDR/RR-TB + SLI & FQ resistance | 0.24 (0.31) | 0.06 [0, 0.42] | 0 to 1 | 4.1 (1.4, 6.9) | 4.6 (0.2, 9.0) |
| MDR/RR-TB + Pyrazinamide resistance | 0.45 (0.30) | 0.44 [0.21, 0.65] | 0 to 1 | 4.3 (0.7, 7.9) | |
| **Drugs used in treatment** | | | | | |
| Patients received Bedaquiline | 0.32 (0.44) | 0 [0, 1] | 0 to 1 | 1.5 (-0.6, 3.5) | |
| Patients received Linezolid | 0.37 (0.40) | 0.23 [0, 0.73] | 0 to 1 | 0.03 (-2.2, 2.2) | |
| Bedaquiline used at site (%) | 41 (48.8) | NE | NE | 0.17 (-1.6, 2.0) | -0.3 (-2.6, 2.0) |
| Linezolid used at site (%) | 58 (69.0) | NE | NE | -0.59 (-2.5, 1.3) | -0.3 (-2.8, 2.2) |
| Patients with success | 79.8 (236.7) | 34.5 [11, 76.5] | 1 to 2128 | -0.03 (-0.12, 0.07) | |
| Patients treated | 131.3 (401.5) | 52.5 [15.8, 123.3] | 1 to 3626 | -0.01 (-0.06, 0.05) | 0.0 (-0.1, 0.1) |
| Treatment duration (Months) | 22.8 (4.1) | 22.6 [20.3, 24.6] | 12 to 36 | NE | NE |

Note: Extensive disease is defined as: AFB smear positive at baseline. If AFB smear information missing, then if radiographic findings of cavitation or bilateral disease. If value blank in multivariable coefficient column, then the variable was not included in the multivariable model. NE: not estimated; TB: tuberculosis; FQ: fluoroquinolones; SLI: second-line injectable. Note: proportion of patients receiving bedaquiline/linezolid, MDR-RR-TB FQ & SLI sensitive, MDR/RR-TB plus pyrazinamide resistance, and number of patients with success were not included in the multivariable model as they were highly correlated with other relevant variables that were included. For the multivariable model, $R^2$: 0.24; adjusted $R^2$: 0.05

conventions, or access to medications are more important determinants of treatment duration. In contrast, several clinical and treatment factors were associated with individual treatment duration in our analysis, which have shown to be associated with treatment outcomes in several prior studies [15, 20, 21, 28]. Hence, our novel approach of using individual deviation from the site-specific mean treatment duration may provide a better method to assess clinical and treatment characteristics association with treatment duration.

By accounting for the mean treatment duration of a site in the duration outcome and by restriction to patients with successful treatment outcomes we aimed to create an outcome variable that accounts for the site-level variation and outcome-dependent complexities inherent in studying duration for treatment of MDR/RR-TB. The finding that factors predicting poor

**Table 2. Description of patient characteristics and their association (adjusted for age and sex) with deviation in treatment duration from site mean.**

| | n (%) unless specified otherwise | Total treatment duration | Individual Deviation in treatment duration from centre mean | |
| --- | --- | --- | --- | --- |
| | | mean (SD) Months | mean (SD) Months | Univariable regression estimate months (95% CI)* |
| All patients **n = 6702** | | 22.0 (4.6) | 0.0 (4) | |
| **Clinical characteristics** | | | | |
| Sex = Male^Δ | 3982 (59.4) | 22.1 (4.6) | 0.1 (4) | Ref |
| Female | 2719 (40.6) | 21.9 (4.6) | -0.1 (4) | -0.18 (-0.37, 0.02) |
| Age (mean (SD)) | 37.02 (13) | NE | NE | 0.01 (-0.003, 0.01)§ |
| Body mass index (mean (SD)) | 20.47 (3.84) | NE | NE | -0.04 (-0.07, -0.01)§ |
| Body mass index category | | | | |
| Normal | 2024 (30.2) | 22.4 (4.8) | 0 (4.2) | Ref |
| Underweight | 1028 (15.3) | 22.7 (4.3) | 0.4 (3.7) | -0.21 (-0.62, 0.19) |
| Overweight/Obese | 377 (5.6) | 22.3 (4.6) | -0.2 (3.9) | 0.26 (0.00, 0.52) |
| Missing | 3273 (48.8) | 21.5 (4.5) | -0.1 (3.9) | Not estimated |
| 2018 World Bank income category | | | | |
| Low/Low-middle | 1226 (18.3) | 22.5 (4.2) | 0 (3.5) | Ref |
| Upper-Middle | 3555 (53.0) | 22.3 (4) | 0 (3.6) | 0.01 (-0.26, 0.26) |
| High | 1921 (28.7) | 21.3 (5.6) | 0 (4.9) | -0.02 (-0.30, 0.27) |
| Smoking | | | | |
| Ex-smoker or never smoker | 1834 (27.4) | 22.5 (5.3) | -0.1 (4.8) | Ref |
| Current smoker | 939 (14.0) | 22.5 (5) | 0.4 (4.1) | 0.17 (-0.09, 0.42) |
| Unknown | 3929 (58.6) | 21.7 (4.1) | -0.1 (3.5) | Not estimated |
| HIV | | | | |
| Negative | 4771 (71.2) | 22 (4.8) | 0 (4.1) | Ref |
| Positive | 1859 (27.7) | 22.1 (3.9) | 0.1 (3.5) | 0.13 (-0.08, 0.35) |
| Unknown | 72 (1.1) | 22.9 (5.4) | -0.1 (5.1) | Not estimated |
| If HIV positive, on ART | 1686 (90.7) | 22 (3.8) | 0 (3.5) | -0.16 (-0.83, 0.50) |
| Not on ART | 173 (9.3) | 23.2 (4.4) | 0.5 (4.1) | |
| Diabetes | | | | |
| No | 3311 (49.4) | 22.4 (5) | 0 (4.3) | Ref |
| Yes | 466 (7.0) | 21.9 (4.4) | 0.3 (3.7) | 0.21 (-0.18, 0.59) |
| Unknown | 2925 (43.6) | 21.7 (4) | 0 (3.6) | Not estimated |
| Cavitation on X-ray | | | | |
| No | 1606 (24.0) | 21.7 (4.9) | -0.4 (4.2) | Ref |
| Yes | 2308 (34.4) | 22.5 (5.1) | 0.3 (4.3) | 0.60 (0.35, 0.86) |
| Unknown | 2788 (41.6) | 21.8 (3.8) | 0 (3.6) | 0.37 (0.12, 0.61) |
| Bilateral disease | | | | |
| No | 1122 (16.7) | 21.4 (4.9) | -0.3 (4) | Ref |
| Yes | 1999 (29.8) | 22.2 (4.9) | 0.2 (4.1) | 0.52 (0.22, 0.81) |
| Unknown | 3581 (53.4) | 22.1 (4.3) | 0 (3.9) | 0.35 (0.09, 0.62) |
| AFB smear result | | | | |
| Neg | 1974 (29.5) | 21.4 (4.7) | -0.6 (4) | Ref |
| Pos | 4280 (63.9) | 22.4 (4.5) | 0.3 (4) | 0.91 (0.70, 1.12) |
| Unknown | 448 (6.7) | 21.2 (4.2) | 0 (3.8) | 0.65 (0.24, 1.05) |
| Extensive disease | | | | |
| No | 2147 (32.0) | 21.3 (4.6) | -0.6 (3.9) | |
| Yes | 4512 (67.8) | 22.4 (4.6) | 0.3 (4) | 0.90 (0.70, 1.11) |
| Unknown | 43 (0.0) | 20.9 (4.7) | -0.1 (4.5) | Not estimated |

*(Continued)*

**Table 2.** (Continued)

| | n (%) unless specified otherwise | Total treatment duration | Individual Deviation in treatment duration from centre mean | |
|---|---|---|---|---|
| | | mean (SD) Months | mean (SD) Months | Univariable regression estimate months (95% CI)* |
| **Treatment history and markers of disease severity** | | | | |
| Past TB treatment | | | | |
| No | 2336 (34.9) | 21.2 (4.4) | -0.4 (3.8) | Ref |
| Yes | 4271 (63.7) | 22.5 (4.6) | 0.2 (4.1) | 0.56 (0.36, 0.77) |
| Unknown | 95 (1.4) | 21.7 (4.5) | 0.1 (3.8) | 0.48 (-0.34, 1.29) |
| Past first-line TB drug use | | | | |
| No | 2336 (34.9) | 21.2 (4.4) | -0.4 (3.8) | Ref |
| Yes | 4271 (63.7) | 22.5 (4.6) | 0.2 (4.1) | 0.56 (0.36, 0.76) |
| Unknown | 95 (1.4) | 21.7 (4.5) | 0.1 (3.8) | Not estimated |
| Past second-line TB drug used | | | | |
| No | 5048 (75.3) | 21.7 (4.1) | -0.2 (3.6) | Ref |
| Yes | 1226 (18.3) | 23.3 (5.4) | 0.6 (4.6) | 0.71 (0.44, 0.98) |
| Unknown | 428 (6.4) | 22.4 (6.2) | 0.5 (5.6) | Not estimated |
| **Pre-treatment Drug susceptibility results** | | | | |
| DST Performed for FQ | 6449 (96.2) | | | Not estimated |
| If DST Performed, FQ Resistant = Yes | 1172 (18.2) | 23.6 (5.8) | 0.8 (5) | 1.04 (0.77, 1.31) |
| If DST Performed, FQ Resistant = No | 5277 (81.8) | 21.6 (4.1) | -0.2 (3.6) | Ref |
| DST Performed for SLIs | 6455 (96.3) | | | Not estimated |
| If DST Performed, SLI Resistant = Yes | 1629 (25.2) | 23 (5.3) | 0.5 (4.5) | 0.58 (0.36, 0.81) |
| If DST Performed, SLI Resistant = No | 4826 (74.8) | 21.7 (4.2) | -0.1 (3.8) | Ref |
| DST Performed for Linezolid | 665 (9.9) | | | Not estimated |
| If DST Performed, Linezolid Resistant = Yes | 16 (2.4) | 21.5 (3.6) | -0.8 (2.8) | -0.76 (-2.74, 1.22) |
| If DST Performed, Linezolid Resistant = No | 649 (97.6) | 21.1 (4.4) | 0 (3.7) | |
| DST Performed for Pyrazinamide | 3490 (52.1) | | | Not estimated |
| If DST Performed, Pyrazinamide Resistant = Yes | 1859 (53.3) | 22.2 (5.4) | 0.3 (4.7) | 0.51 (0.30, 0.72) |
| If DST Performed, Pyrazinamide Resistant = No | 1631 (46.7) | 21.1 (4.6) | -0.4 (4) | Ref |
| DST Performed for Clofazimine | 252 (3.8) | | | Not estimated |
| If DST Performed, Clofazimine Resistant = Yes | 9 (3.6) | 24.4 (5) | 2 (5) | Not estimated † |
| If DST Performed, Clofazimine Resistant = No | 243 (96.4) | 21.8 (5.8) | 0.1 (4.4) | |
| DST Performed for Cycloserine‡ | 2034 (30.3) | | | Not estimated |
| If DST Performed, Cycloserine Resistant = Yes | 260 (12.8) | 23.4 (6.2) | 1 (5.2) | 1.16 (0.65, 1.68) |
| If DST Performed, Cycloserine Resistant = No | 1774 (87.2) | 21.9 (5.3) | -0.1 (4.5) | |
| MDR category | | | | |
| MDR/RR-TB FQ &SLI sensitive | 4337 (64.7) | 21.5 (4) | -0.3 (3.5) | Ref |
| MDR/RR-TB + FQ resistant & SLI sensitive | 929 (13.9) | 22.2 (4.5) | 0.2 (3.9) | 0.48 (0.20, 0.77) |
| MDR/RR-TB + SLI resistant & FQ sensitive | 475 (7.1) | 23.2 (5.8) | 0.9 (5.1) | 1.24 (0.85, 1.63) |
| MDR/RR-TB + SLI & FQ resistance | 688 (10.3) | 23.9 (5.9) | 0.8 (5) | 1.08 (0.76, 1.41) |

(*Continued*)

**Table 2.** (Continued)

| | n (%) unless specified otherwise | Total treatment duration | Individual Deviation in treatment duration from centre mean | |
|---|---|---|---|---|
| | | mean (SD) Months | mean (SD) Months | Univariable regression estimate months (95% CI)* |
| No DST | 273 (4.1) | 22.5 (5.7) | 0 (5) | Not estimated |
| MDR/RR-TB + SLI & FQ resistance vs. all others | | | | |
| No | 5741 (85.7) | 21.8 (4.3) | -0.1 (3.8) | Ref |
| Yes | 688 (10.3) | 23.9 (5.9) | 0.8 (5) | 0.83 (0.52, 1.14) |
| Unknown | 273 (4.1) | 22.5 (5.7) | 0 (5) | Not estimated |

treatment outcomes such as MDR with additional resistance to FQ and/or SLI [1, 3], HIV infection [20], or cavitation [11, 29] were associated with longer treatment duration provides support for the use of this method. Our finding that treatment duration is shorter when bedaquiline was used, is supported by several studies that have established the efficacy of bedaquiline [3, 15, 28], which further supports the use of bedaquiline containing regimens for all MDR/RR-TB patients. Additionally, as this was observational data, we included patient populations in our analysis that were excluded from trials of shorter treatment, such as those with additional resistance to SLIs or FQs [9, 10], low body mass index [11, 29], low HIV CD4 cell counts [9–11, 29], "any comorbidity likely to compromise protocol assessments" [11, 29], or extensive disease and past treatment (the last two groups are not eligible for the 9-month all-oral regimen in WHO guidelines [3]). Use of bedaquiline was associated with shorter treatment duration across the majority of subgroups, suggesting that inclusion of patients previously excluded from RCTs [9–11, 29] or considered ineligible for short MDR treatment in guidelines [3] could be included in future trials of shorter bedaquiline and/or pretomanid containing regimens (such as bedaquiline, pretomanid, & linezolid (BPaL), or BPaL plus moxifloxacin, BPaLM) most recently recommended by the WHO [3]. Additionally, our results indicate that certain patients with more complicated clinical profiles, such as MDR/RR-TB patients without additional resistance who also have extensive disease but either no past treatment or no HIV, may benefit from shorter treatment.

Although our analysis indicated that use of bedaquiline was associated with shorter treatment duration, these results require cautious interpretation as our models were constructed for the purposes of hypothesis generation. The association of shorter duration with use of bedaquiline may reflect the preferred use of the drug in regimens with planned shorter durations. However, this was not observed with linezolid or FQs, which are also used in regimens with shorter planned durations. Additionally, we assessed drugs that were received at any time (ever) during treatment, which does not adequately account for regimen changes. Some characteristics and drugs that were associated with longer duration may reflect clinical conventions. For instance, use of low efficacy drugs (e.g. clarithromycin, Amx-Clv, and injectables [15]) may reflect use of drugs in desperation for patients with more complicated disease with longer planned duration. Similar conventions apply to associations with cavitation.

Our study has limitations. Primarily, we conducted this analysis in a population treated between 1993 and 2019, and treatment practises, including use of fluoroquinolones, bedaquiline, and second-line injectables, as well as advancement in antiretroviral therapy (ART) and their uptake, have changed substantially in the last five years [3]. We also did not have data on the number of cavities, only the presence or absence, nor did we have data on level of AFB smear positivity (only positive or negative), and were unable to assess what effect this had on

**Table 3. Description of drugs used in treatment and their association (adjusted for age and sex) with deviation in treatment duration from site mean.**

| All patients n = 6702 | n (%) unless specified otherwise | Total treatment duration | Individual Deviation in treatment duration from centre mean | |
|---|---|---|---|---|
| | | mean (SD) Months | mean (SD) Months | Univariable regression estimate months (95% CI)* |
| **Drugs used in treatment** | | | | |
| Used Bedaquiline Ever During Treatment = Yes | 1605 (23.9) | 22.4 (3.9) | 0.2 (3.5) | 0.27 (0.04, 0.49) |
| No | 5097 (76.1) | 21.9 (4.8) | -0.1 (4.1) | Ref |
| Used Ofloxacin Ever During Treatment = Yes | 1373 (20.5) | 22 (4.2) | -0.1 (3.6) | -0.13 (-0.36, 0.11) |
| No | 5329 (79.5) | 22 (4.7) | 0 (4.1) | Ref |
| Used Ciprofloxacin Ever During Treatment = Yes | 266 (4.0) | 23 (5.8) | 0 (5.4) | 0.03 (-0.46, 0.52) |
| No | 6436 (96.0) | 22 (4.5) | 0 (3.9) | Ref |
| Used Moxifloxacin Ever During Treatment = Yes | 3459 (51.6) | 22.1 (4.6) | 0.2 (4.1) | 0.41 (0.22, 0.60) |
| No | 3243 (48.4) | 21.9 (4.6) | -0.2 (3.9) | Ref |
| Used Levofloxacin Ever During Treatment = Yes | 1889 (28.2) | 21.8 (4.9) | 0.1 (4.2) | 0.19 (-0.03, 0.40) |
| No | 4813 (71.8) | 22.1 (4.4) | -0.1 (3.9) | Ref |
| Used Linezolid Ever During Treatment = Yes | 1594 (23.8) | 22.5 (5.1) | 0.5 (4.4) | 0.63 (0.41, 0.86) |
| No | 5108 (76.2) | 21.9 (4.4) | -0.1 (3.8) | Ref |
| Used Clofazimine Ever During Treatment = Yes | 1101 (16.4) | 22.5 (4.8) | 0.3 (3.8) | 0.35 (0.10, 0.61) |
| No | 5601 (83.6) | 21.9 (4.5) | -0.1 (4) | Ref |
| Used Cycloserine/Terizidone Ever During Treatment = Yes | 5702 (85.1) | 22.1 (4.4) | 0 (4.0) | 0.19 (-0.08, 0.46) |
| No | 1000 (14.9) | 21.7 (5.3) | -0.2 (4.1) | Ref |
| Used Ethambutol Ever During Treatment = Yes | 2895 (43.2) | 22 (4.5) | 0.1 (3.9) | 0.10 (-0.09, 0.29) |
| No | 3807 (56.8) | 22.1 (4.7) | 0 (4.1) | Ref |
| Used Pyrazinamide Ever During Treatment = Yes | 5175 (77.2) | 22 (4.3) | 0 (3.8) | -0.13 (-0.36, 0.09) |
| No | 1527 (22.8) | 22 (5.5) | 0.1 (4.6) | Ref |
| Used Streptomycin Ever During Treatment = Yes | 692 (10.3) | 22.5 (5.1) | 0.1 (4.5) | 0.10 (-0.21, 0.42) |
| No | 6010 (89.7) | 22 (4.5) | 0 (3.9) | Ref |
| Used Rifabutin Ever During Treatment = Yes | 154 (2.3) | 22.8 (6.7) | 0.1 (5.8) | 0.05 (-0.59, 0.69) |
| No | 6548 (97.7) | 22 (4.5) | 0 (3.9) | Ref |
| Used Amikacin Ever During Treatment = Yes | 1048 (15.6) | 21.8 (4.8) | 0 (4.2) | 0.06 (-0.21, 0.32) |
| No | 5654 (84.4) | 22.1 (4.5) | 0 (3.9) | Ref |
| Used Capreomycin Ever During Treatment = Yes | 1446 (21.6) | 23.1 (5.6) | 0.5 (4.7) | 0.66 (0.42, 0.89) |
| No | 5256 (78.4) | 21.7 (4.2) | -0.1 (3.8) | Ref |
| Used Kanamycin Ever During Treatment = Yes | 3151 (47.0) | 21.9 (3.9) | 0.1 (3.5) | 0.20 (0.01, 0.39) |
| No | 3551 (53.0) | 22.1 (5.1) | -0.1 (4.3) | Ref |
| Used Ethionamide/Prothionamide Ever During Treatment = Yes | 5096 (76.0) | 22.1 (4.5) | 0 (4) | 0..20 (-0.03, 0.42) |
| No | 1606 (24.0) | 21.8 (4.9) | -0.2 (4.1) | Ref |
| Used PAS Ever During Treatment = Yes | 2759 (41.2) | 22.7 (5.2) | 0.3 (4.4) | 0.48 (0.29, 0.68) |
| No | 3943 (58.8) | 21.6 (4.1) | -0.2 (3.6) | Ref |
| Used Amx-Clv Ever During Treatment = Yes | 994 (14.8) | 24 (6.2) | 1 (5.5) | 1.27 (0.99, 1.55) |
| No | 5708 (85.2) | 21.7 (4.1) | -0.2 (3.6) | Ref |
| Used Thioacetazone Ever During Treatment = Yes | 68 (1.0) | 21 (5.4) | 0.2 (4) | 0.20 (-0.75, 1.16) |
| No | 6634 (99.0) | 22 (4.6) | 0 (4) | Ref |
| Used Clarithromycin Ever During Treatment = Yes | 485 (7.2) | 24.6 (7) | 1.4 (6.2) | 1.50 (1.14, 1.87) |
| No | 6217 (92.8) | 21.8 (4.3) | -0.1 (3.7) | Ref |
| Used Imipenem Ever During Treatment = Yes | 237 (3.5) | 23.4 (4.6) | 0.4 (4.1) | 0.37 (-0.14, 0.89) |
| No | 6465 (96.5) | 22 (4.6) | 0 (4) | Ref |

*(Continued)*

**Table 3.** (Continued)

| All patients n = 6702 | n (%) unless specified otherwise | Total treatment duration | Individual Deviation in treatment duration from centre mean | |
|---|---|---|---|---|
| | | mean (SD) Months | mean (SD) Months | Univariable regression estimate months (95% CI)* |
| Used Meropenem Ever During Treatment = Yes | 61 (0.9) | 21.1 (5.4) | -0.3 (4.4) | -0.32 (-1.33, 0.68) |
| No | 6641 (99.1) | 22 (4.6) | 0 (4) | Ref |
| Used Delamanid Ever During Treatment = Yes | 114 (1.7) | 21 (4.4) | 0 (3.6) | -0.02 (-0.76, 0.72) |
| No | 6588 (98.3) | 22 (4.6) | 0 (4) | Ref |
| Number of drugs (median [IQR]) | 5 [4, 6] | NE | NE | 0.37 (0.29, 0.46)§ |
| Number of effective drugs (median [IQR]) | 4 [4, 5] | NE | NE | 0.03 (-0.06, 0.12)§ |
| Number of limited access drugs** (median [IQR]) | 0 [0, 1] | NE | NE | 0.18 (0.09, 0.26)§ |
| Total treatment duration (median [IQR]) | 22 [19, 24] | NE | NE | Not estimated |
| Deviation in treatment duration (median [IQR]) | -0.15 [-2, 2] | NE | NE | Not estimated |

SD: standard deviation; XDR: extensively drug resistant tuberculosis; MDR: multidrug resistant tuberculosis; TB: tuberculosis; AFB: acid-fast bacillus; Amx-Clv: Amoxicillin-Clavulanic Acid

* Regression coefficients were estimated using imputed data and adjusted for age and sex.

† Too few observations to estimate.

‡ Drug susceptibility testing for cycloserine and terizidone combined.

§ per unit increase.

** includes bedaquiline, clofazimine, linezolid, imipenem, and meropenem.

Δ One subject missing sex.

duration. Further, studies conducted exclusively in children were excluded from the IPD, and we were not able to assess associations in this population. Additionally, there are site-level differences in treatment protocols that may affect treatment outcomes (availability of drugs), that may not be captured in variables we included in our models. However, as we used the average duration of treatment at a site in our duration outcome, we believe this may account for site-level heterogeneity in clinical practice. There is still potential for indication bias affecting duration of treatment in patients with complex profiles which may not be accounted for with our outcome. Although we conducted a 'new user' subgroup analysis [30] in those without previous treatment, this only addresses one aspect that may create indication bias for treatment duration. Because our population included only those with treatment success, our findings may not be generalizable to all patients with MDR/RR-TB, although we conducted inverse probability of selection weighted analyses which indicated no substantial differences in the excluded vs included populations. However, adverse events may be more common with longer treatment, and contribute to treatment failure (and as such were excluded in this study) yet it is possible that patients with adverse events would benefit even more from shorter treatment. Finally, our results do not reflect causal relationships and should be interpreted with caution, with additional consideration that variance of regression estimates may be underestimated due to the statistical selection of variables [31].

Despite that, our study has several strengths. We included a large population of patients who had detailed information on important clinical characteristics and treatment. Additionally, we conducted several subgroup analyses of important patient groups that were previously excluded from trials on shorter treatment [9–11, 29]. We also used E-values to assess unmeasured confounding (where a larger E-values implies a more robust observed estimate). Although plausible that an unmeasured confounder could account for the observed association with bedaquiline, we feel it is unlikely such an important predictor would not have been

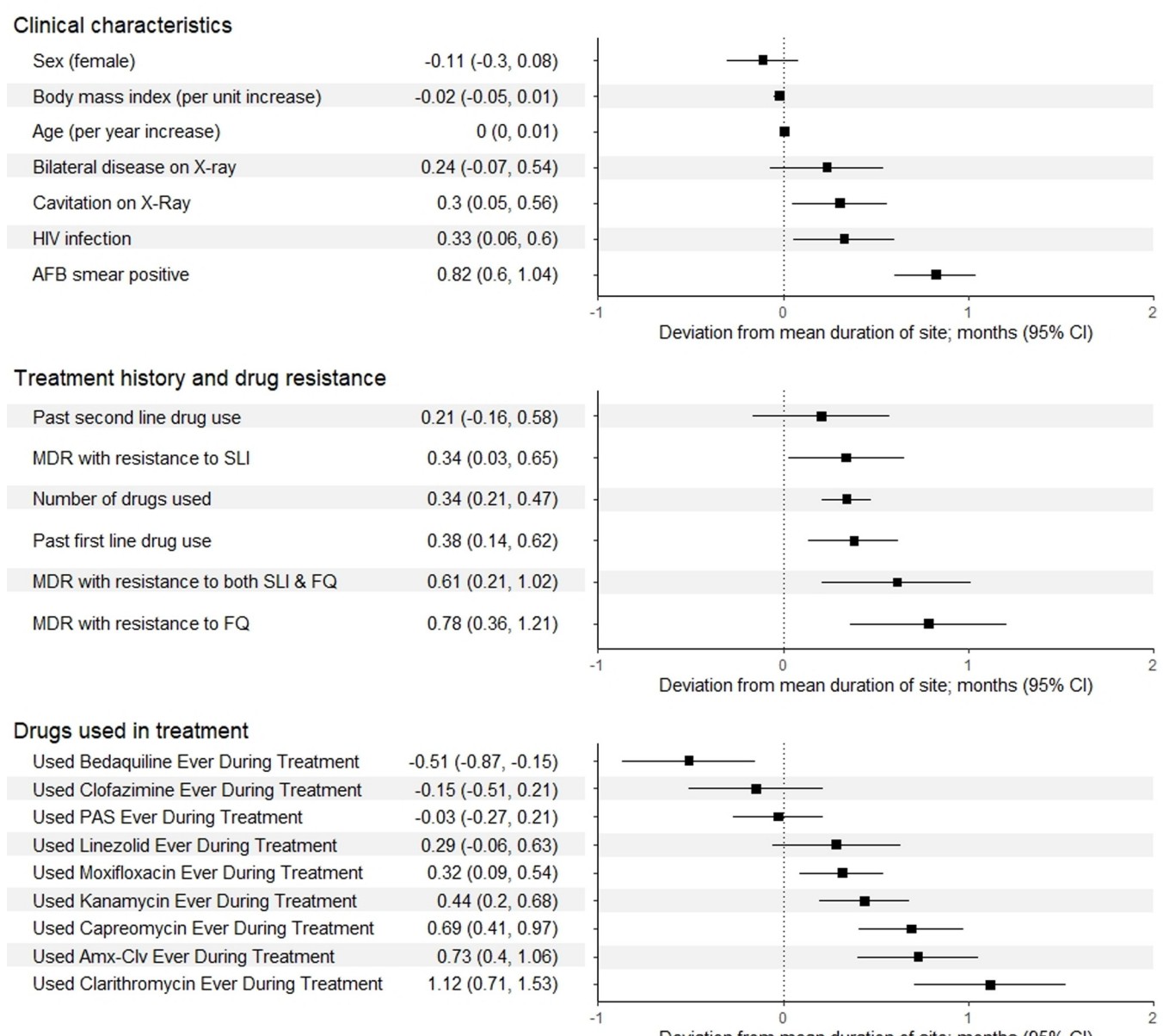

**Fig 2. Forest plot of associations between deviation in treatment duration (in months) from site mean and patient characteristics, resistance categories.** Estimates and 95% confidence intervals (CI) from a multivariable linear mixed model including all variables shown. Footnote: * Conditional R2 for model: 0.08.

included in our data. Although RCTs can provide clearer evidence on optimal duration, these are expensive, time consuming, often lack generalizability, and can test only a limited number of durations and/or regimens at once. As well, evidence from cohorts are useful as potential indicators of success of these regimens in programmatic settings [32, 33]). Our use of observational data from a large population of MDR/RR-TB patients from 84 treatment sites and 34 countries provided evidence that should be more generalizable. We also describe an analysis of characteristics associated with duration at the level of the treatment site. We interpret that the lack of associations between patient or treatment characteristics and our outcome may reflect

**Table 4. Associations of individual deviation in treatment duration from site mean with patient characteristics, resistance categories, and drugs used, within specified subgroups.** Estimates and 95% confidence interval (CI) from multivariable linear mixed models including all variables shown (unless otherwise specified).

| | Patients with additional SLI & FQ resistance | Patients without FQ and SLI resistance* | Patients with extensive disease | Patients without extensive disease | Patients with past TB treatment | Patients without past TB treatment |
|---|---|---|---|---|---|---|
| Characteristic | months (95% CI) | months (95% CI) | months (95% CI) | months (95% CI) | months (95% CI) | months (95% CI) |
| **Clinical characteristics** | | | | | | |
| Age (per year increase) | 0.02 (-0.01, 0.05) | 0 (0, 0.01) | 0 (-0.01, 0.01) | 0.01 (0, 0.03) | 0.01 (0, 0.02) | 0 (-0.02, 0.01) |
| Sex (Female) | 0.16 (-0.6, 0.92) | -0.14 (-0.34, 0.06) | -0.26–0.5, -0.02) | 0.12 (-0,21, 0.46) | -0.08 (-0.32, 0.17) | -0.17 (-0.48, 0.14) |
| Body mass index (per unit increase) | 0.01 (-0.1, 0.11) | -0.03 (-0.06, 0.01) | -0.02 (-0.06, 0.02) | -0.03 (-0.09, 0.02) | -0.03 (-0.07, 0.01) | -0.01 (-0.07, 0.05) |
| HIV infection | 0.15 (-1.01, 1.31) | 0.34 (0.06, 0.62) | 0.33 (0.01, 0.66) | 0.33 (-0.1, 0.76) | 0.38 (0.02, 0.74) | 0.22 (-0.18, 0.62) |
| AFB smear positive | 0.77 (-0.08, 1.62) | 0.79 (0.56, 1.01) | Not estimated | Not estimated | 0.82 (0.52, 1.12) | 0.77 (0.44, 1.1) |
| Cavitation on X-Ray | 1.08 (0.1, 2.05) | 0.22 (-0.05, 0.49) | Not estimated | Not estimated | 0.23 (-0.09, 0.54) | 0.59 (0.12, 1.06) |
| Bilateral disease on X-ray | 0.64 (-0.56, 1.85) | 0.16 (-0.16, 0.48) | Not estimated | Not estimated | 0.36 (-0.01, 0.74) | -0.04 (-0.58, 0.5) |
| **Treatment history and drug resistance** | | | | | | |
| Past first-line drug use | -0.22 (-1.56, 1.11) | 0.4 (0.16, 0.65) | 0.43 (0.13, 0.74) | 0.34 (-0.05, 0.73) | Not estimated | Not estimated |
| Past second-line drug use | 0.72 (-0.41, 1.86) | 0.15 (-0.24, 0.55) | 0.13 (-0.32, 0.57) | 0.4 (-0.16, 0.97) | Not estimated | Not estimated |
| Number of drugs used (per unit increase) | 0.35 (-0.09, 0.79) | 0.33 (0.19, 0.47) | 0.4 (0.24, 0.56) | 0.16 (-0.06, 0.38) | 0.31 (0.14, 0.47) | 0.29 (0.07, 0.51) |
| MDR/RR-TB + FQ resistant & SLI sensitive | Not estimated | Not estimated | 0.59 (0.21, 0.98) | -0.2 (-0.75, 0.34) | 0.12 (-0.29, 0.54) | 0.62 (0.15, 1.08) |
| MDR/RR-TB + SLI resistant & FQ sensitive | Not estimated | Not estimated | 0.92 (0.38, 1.45) | 0.46 (-0.24, 1.15) | 0.75 (0.26, 1.24) | 0.97 (0.1, 1.83) |
| MDR/RR-TB + SLI & FQ resistance | Not estimated | Not estimated | 0.84 (0.32, 1.36) | 0.15 (-0.52, 0.81) | 0.55 (0.08, 1.02) | 0.86 (0.04, 1.68) |
| **Drugs used in treatment** | | | | | | |
| Used Bedaquiline Ever During Treatment | -0.89 (-2.19, 0.41) | -0.47 (-0.86, -0.09) | -0.51 (-0.94, -0.09) | -0.61 (-1.25, 0.03) | -0.43 (-0.89, 0.03) | -0.52 (-1.09, 0.05) |
| Used Moxifloxacin Ever During Treatment | 0.63 (-0.2, 1.47) | 0.26 (0.02, 0.5) | 0.42 (0.14, 0.7) | 0.07 (-0.31, 0.45) | 0.18 (-0.1, 0.46) | 0.7 (0.31, 1.09) |
| Used Linezolid Ever During Treatment | -0.82 (-2.03, 0.38) | 0.54 (0.15, 0.93) | 0.22 (-0.19, 0.64) | 0.44 (-0.18, 1.06) | 0.12 (-0.32, 0.56) | 0.69 (0.12, 1.27) |
| Used Clofazimine Ever During Treatment | 0.54 (-0.51, 1.58) | -0.43 (-0.84, -0.02) | -0.37 (-0.8, 0.07) | 0.3 (-0.31, 0.91) | 0.01 (-0.45, 0.47) | -0.25 (-0.85, 0.34) |
| Used Capreomycin During Treatment | 0.46 (-0.44, 1.36) | 0.8 (0.49, 1.11) | 0.74 (0.4, 1.09) | 0.56 (0.08, 1.04) | 0.75 (0.39, 1.1) | 0.73 (0.25, 1.22) |
| Used Kanamycin Ever During Treatment | 0.64 (-0.58, 1.86) | 0.47 (0.22, 0.71) | 0.48 (0.18, 0.77) | 0.45 (0.02, 0.88) | 0.21 (-0.09, 0.52) | 0.9 (0.48, 1.31) |
| Used Amx-Clv Ever During Treatment | 0.7 (-0.18, 1.59) | 0.77 (0.39, 1.15) | 0.84 (0.43, 1.25) | 0.51 (-0.03, 1.05) | 0.93 (0.53, 1.33) | 0.43 (-0.16, 1.03) |
| Used Clarithromycin During Treatment | 0.64 (-0.46, 1.73) | 1.19 (0.73, 1.66) | 0.88 (0.38, 1.39) | 1.6 (0.9, 2.3) | 1.25 (0.77, 1.73) | 0.72 (-0.1, 1.55) |

*Includes MDR/RR-TB FQ &SLI sensitive, MDR/RR-TB + FQ resistant & SLI sensitive, MDR/RR-TB + SLI resistant & FQ sensitive. Note: for MDR models were also adjusted for resistance to fluoroquinolone (FQ), second line injectables (SLI), pyrazinamide and cycloserine. For MDR/RR-TB + SLI & FQ resistance, models were also adjusted for resistance to pyrazinamide and cycloserine (not shown for consistency with other subgroups). All models also adjusted for use of PAS. MDR: multidrug resistant tuberculosis; TB: tuberculosis; AFB: acid-fast bacillus; Amx-Clv: Amoxicillin-Clavulinic Acid

the impact that provider belief and site convention have on duration, a problem which has not been previously described.

Our results produced correlates of individual treatment duration in MDR/RR-TB patients that may help identify patients who would benefit from shorter treatment. We found evidence

that certain patients with more extensive disease and drug resistance may benefit from shorter treatment and could be included in future treatment shortening trials.

## Supporting information

**S1 File.**
(DOCX)

## Author Contributions

**Conceptualization:** Nicholas Winters, Dick Menzies.

**Formal analysis:** Nicholas Winters.

**Methodology:** Nicholas Winters, Mireille E. Schnitzer, Jonathon R. Campbell, Susannah Ripley, Rada Savic, Nafees Ahmad, Dick Menzies.

**Validation:** Nicholas Winters.

**Writing – original draft:** Nicholas Winters.

**Writing – review & editing:** Nicholas Winters, Mireille E. Schnitzer, Jonathon R. Campbell, Susannah Ripley, Carla Winston, Rada Savic, Nafees Ahmad, Gregory Bisson, Keertan Dheda, Ali Esmail, Medea Gegia, Ignacio Monedero, Margareth Pretti Dalcolmo, Denise Rodrigues, Rupak Singla, Jae-Joon Yim, Dick Menzies.

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
