## [Decision Letter · Decision Letter 0]

7 Aug 2023

PONE-D-23-17278Identifying patients with multidrug-resistant tuberculosis who may benefit from shorter durations of treatment.PLOS ONE

Dear Dr. Winters,

Thank you for submitting your manuscript to PLOS ONE. After careful consideration, we feel that it has merit but does not fully meet PLOS ONE’s publication criteria as it currently stands. Therefore, we invite you to submit a revised version of the manuscript that addresses the points raised during the review process. Please see the comments from reviewers, below.

We look forward to receiving your revised manuscript.

Kind regards,

Dzintars Gotham

Academic Editor

PLOS ONE

Reviewers' comments:

Reviewer's Responses to Questions

**Comments to the Author**

1. Is the manuscript technically sound, and do the data support the conclusions?

Reviewer #1: Partly

Reviewer #2: Yes

Reviewer #3: Yes

2. Has the statistical analysis been performed appropriately and rigorously? 

Reviewer #1: Yes

Reviewer #2: Yes

Reviewer #3: Yes

3. Have the authors made all data underlying the findings in their manuscript fully available?

Reviewer #1: Yes

Reviewer #2: Yes

Reviewer #3: Yes

4. Is the manuscript presented in an intelligible fashion and written in standard English?

Reviewer #1: Yes

Reviewer #2: Yes

Reviewer #3: Yes

5. Review Comments to the Author

Reviewer #1: There has been considerable work done by the research team working with large data sets with the methods well articulated.

The data sets included cover a large date range for the studies (between 1993 and 2019) when there has been considerable changes in WHO recommended treatments and diagnosis for MDRTB, including the use of rapid molecular diagnostic tests for diagnosis of Rif Resistant TB (Xpert), the introduction of Bedaquilline in 2013 and the 9 month regimen in 2016. The article would benefit in highlighting the timelines of key changes in WHO guidance with regards to MDRTB treatment to help orientate the reader - potential a figure/timeline of guidance with regards to priority/group A drugs, recommended durations and introduction of new drugs. If the majority of the earlier studies were excluded in the process outlined in Fig 1 that would be interesting information to know given the progression of the field since 1993. Did the researchers see any impact of the guidance of a 9 month regimen in 2016 on the mean duration of treatment between 2016 - 2019 as that was the first time that WHO recommended a duration of treatment less than 18months?

For the definition used for extensive disease (line 164) the criteria for CXR matches the WHO definition (Presence of bilateral cavitary disease or extensive parenchymal damage on CXR) but the use of the broad statement of AFB positivity encompasses everything from scanty to 3+ which have different impacts on severity of disease. Did the data sets give the level of smear AFB positivity? If this was given, was scanty/+1 AFB have different durations to 3+ smears? If this is not known, the research team may want to reflect on if this may have had any effect.

Line 210 -214 - "Use of capreomycin, kanamycin, moxifloxacin, levofloxacin, PAS, linezolid, clofazimine, Amx-Clv, clarithromycin, or bedaquiline, as well as greater number of drugs, were all associated with longer treatment duration." It is not clear what this sentence is saying - if any of the these drugs were individually involved in the regimen then the duration was longer? Most of these drugs are key components of DRTB regimens. This statement also seems to conflict with line 217 "Use of bedaquiline was associated with shorter treatment duration by -0.51 (95% CI -0.87 to -0.15) months".

For the statement on line 219 "... and with greater number of drugs used, or use of moxifloxacin, kanamycin, capreomycin, or Amx-Clv." given that prior to the introduction of Bdq in 2013, FLQ and SLI (moxi, Kanamycin and capreomycin) were the backbone of any DRTB regimen so is this saying that the use of these more effective drugs were associated with longer durations?

In the discussion section the researchers do not mention the latest WHO DRTB guidance (Dec 2022) on the 6 month regimens - BPaLM/BPaL. What does this recommendation for a standard 6 month regimen for BPaLM for adults >14yrs with MDR/RR-TB or with MDR/RR-TB and resistance to fluoroquinolones (pre-XDR-TB), regardless of HIV status mean for the approaches outlined in the article? The BPaLM recommendation does restrict the use of this regimen in those with prior exposure to drugs in the regimen (>1month) so the points made regarding these groups may still be valid but it would be good to see the explicit mention of this dramatic advance in shorter effective treatment for DRTB.

Reviewer #2: The authors have taken a novel approach to try to identify factors associated with treatment duration for MDR/RR-TB. This is a hypothesis generating study. The study is well-written, clearly described with appropriate analyses.

Major comments:

As the data involved in the study are from before 2020, and there is no mention in the article of the 6 month all oral regimens, the article doesn’t really address how the approach might be useful in the future for assessing duration – especially if treatment is given as fixed durations based on regimen.

In the supplement 1 under search criteria it states “Studies exclusively in children or of patients treated with short regimens were excluded as these were the topics of two concurrent individual patient data meta-analyses at time of original publication”. If these are indeed excluded it would be helpful to have this stated in the methods. Also then including within the discussion the potential impact or not on the results of this for this methodology.

Minor comments:

Introduction:

the guidelines for treatment referenced are older guidelines. Suggest reference up to date guidelines from 2022. Line 74-75 states “Current recommended treatment for advanced and extensive MDR-TB is as long as 18-20 months”. This is no longer the case with the 2022 current guidelines. The recommendation that “patients without extensive TB disease and without severe extrapulmonary TB” should not have shorter regimen applies to the 9-month regimens from the 2019 guideline. Suggest rephrase this sentence to clarify how things apply with the current guidelines.

Methods

Line 146 spell out abbreviations at first use: 95% confidence interval (95% CI), and line 163 fluoroquinolone (FQ) and second line injectables (SLI), and 288 ART

Suggest state more clearly whether when referring to MDR-TB you are being inclusive (MDR/RR-TB) or only including those with both isoniazid and rifampicin resistance. From line 95-96 it suggests that also includes rifampicin resistance, however in Methods only ever state MDR-TB which is defined as resistance to both rifampicin and isoniazid. In line 100 it refers to study dataset described in reference 14 – would be easier if state clearly in methods that actually inclusion criteria was rifampicin resistant TB (RR-TB). Would suggest either changing to using RR-TB throughout or MDR/RR-TB throughout and only use MDR-TB when specifically meaning resistance as per definition in in lines 71-72.

For the treatment outcomes, 2 references (21 and 22) are mentioned for definitions. These references do have some differences. It would be helpful to the reader to state clearly here whether outcomes given by each site were taken as the outcomes or whether a single set of definitions was retrospectively applied to all date. Also reference 22 is from 2022, while the IPD data was only up til 2019, so not sure if the definitions referred to here were used at all.

Reviewer #3: Winters et al present an analysis of ecological and individual factors associated with successful shorter treatment of patients with MDR TB. Leveraging data from a previously published IPD meta analysis, the authors evaluated data of 6702 patients from 34 different countries. In a site level, the proportion of resistance to FQ-SILI was associated with longer duration, while in at an individual level an adjusted model identified AFB smear, presence of cavities, HIV status and added resistance as factors associated with longer treatment duration. Importantly, the use of bedaquiline was associated with shorter treatment duration. The study is well written and the statistics are appropriate, the topic is important, some comments below:

1. My main concern is that the study relies on data obtained between 2009 and 2018, at that time longer 15-24 month treatment regimens were recommended by WHO. As the authors are aware, current WHO recommendations endorse the use of shorter 6 month all oral regimens using BPaL/BPaLM thus the data presented here albeit interesting may not be applicable to the current MDR TB scenario. The authors should make a point in the discussion into what settings would your data be applicable? Maybe in patients who fail BPaL/BPaLM? In countries without access to pretomanid or bedaquiline? Or in populations were BPaL is not validated yet such as children, pregnancy, end stage renal disease among others.

2. Line 77-79 “In the past 10 years, several studies have investigated shorter regimens for treatment “ of MDR-TB in randomized controlled trials (RCTs), but these may not reflect treatment in programmatic settings” This statement is not fully accurate, there is recent data on the applicability of shorter regimens under programmatic conditions (Haley et al pmid: 37249079, Acuna-Villaorduna et al pmid: 37491751). In the US, outcomes of all oral shorter regimens report > 95% treatment success.

3. The authors included patients with successful treatment outcomes in order to avoid bias which is a reasonable approach, however it could have introduced selection bias. Some of the treatment failure patients who were excluded could have been due to medication side effects and in fact these group may benefit even more of shorter regimens. In fact, in the TB PRACTECAL study, side effects were the main cause (49%) of treatment discontinuation. The authors could consider adding this as a potential limitation

4. The authors could expand more on the outcome definition: successful treatment. They refer to the WHO 2022 document (ref 22) who is more an update on shorter regimens and briefly discuss treatment outcomes definition but specifically for all oral shorter regimens which are not the ones the authors evaluate in their analysis. I suggest the authors to add in the text the precise definition of successful outcome they used (cure (negative cultures) without relapse if evaluated at a given time for instance) and how they managed the differences in outcome definitions between different studies in the IPD datasets.

5. The data on bedaquiline is interesting, I would emphasize more on this. It is not surprising to me that bedaquiline is associated with shorter duration, bedaquiline is highly bactericidal against M. tuberculosis (Yamada et al pmid 36165631) and in animal models can eradicate MTB in 3-4 weeks when used in combination with pretomanid and moxifloxacin. I think the authors could emphasize that their data support the use of bedaquiline for all MDR TB patients.

6. SILI abbreviation (line 164) could be defined, I believe it is second line injectables

6. PLOS authors have the option to publish the peer review history of their article (what does this mean?). If published, this will include your full peer review and any attached files.

Reviewer #1: No

Reviewer #2: No

Reviewer #3: **Yes: **Carlos Acuna-Villaorduna

---

## [Author Response · Author response to Decision Letter 0]

19 Aug 2023

Dear Dr. Gotham,

Thanks for your email of Aug 7, 2023.

We have revised the manuscript as suggested by yourself and the external reviewers, and are re-submitting this manuscript. As requested below we have included a marked-up copy of text with tables embedded, plus a clean copy of same, as well as a point-by-point response to all comments, and a new figure presenting a timeline of changes in WHO recommendations.

Please note that we have included an updated Supplemental file as well. 

We hope this is now acceptable for publication in PLOS One and look forward to hearing from you.

N Winters and D Menzies, on behalf of all authors

Dear Dr. Winters,

Thank you for submitting your manuscript to PLOS ONE. After careful consideration, we feel that it has merit but does not fully meet PLOS ONE’s publication criteria as it currently stands. Therefore, we invite you to submit a revised version of the manuscript that addresses the points raised during the review process.

Please see the comments from reviewers, below.

We look forward to receiving your revised manuscript.

Kind regards,

Dzintars Gotham

Academic Editor

PLOS ONE

Response: the paper has been formatted to meet the above outlined requirements, tables appear after they are cited within the text.

Response: Our ethics statement and participant consent has been included in the methods.

Response: The data is accessible through University College London who now hold the data as a publicly accessible repository. Access to this is governed by an oversight committee, and applications must be made to that committee. There are legal restrictions in terms of signed data sharing agreements with all data contributors. So, application must be made to access the data. But it is publicly accessible and inquiries can be made through this link: https://www.ucl.ac.uk/global-health/research/tb-ipd-platform

Response: Our ethics statement and participant consent has been included in the methods.

Response: We have included a list of captions for our supporting material at the end of the manuscript.

Reviewers' comments:

Reviewer's Responses to Questions

Comments to the Author

1. Is the manuscript technically sound, and do the data support the conclusions?

Reviewer #1: Partly

Reviewer #2: Yes

Reviewer #3: Yes

2. Has the statistical analysis been performed appropriately and rigorously? 

Reviewer #1: Yes

Reviewer #2: Yes

Reviewer #3: Yes

3. Have the authors made all data underlying the findings in their manuscript fully available?

Reviewer #1: Yes

Reviewer #2: Yes

Reviewer #3: Yes

4. Is the manuscript presented in an intelligible fashion and written in standard English?

Reviewer #1: Yes

Reviewer #2: Yes

Reviewer #3: Yes

5. Review Comments to the Author

Reviewer #1: There has been considerable work done by the research team working with large data sets with the methods well articulated. The data sets included cover a large date range for the studies (between 1993 and 2019) when there has been considerable changes in WHO recommended treatments and diagnosis for MDRTB, including the use of rapid molecular diagnostic tests for diagnosis of Rif Resistant TB (Xpert), the introduction of Bedaquilline in 2013 and the 9 month regimen in 2016. The article would benefit in highlighting the timelines of key changes in WHO guidance with regards to MDRTB treatment to help orientate the reader - potential a figure/timeline of guidance with regards to priority/group A drugs, recommended durations and introduction of new drugs. If the majority of the earlier studies were excluded in the process outlined in Fig 1 that would be interesting information to know given the progression of the field since 1993. 

Response: We thank the reviewer for their comment, and we agree that a figure outlining key changes would be helpful, which we have added to the supplement (also see below)

See lines 126 to 127: “(Supplemental Figure S1 presents a timeline for important changes in WHO treatment guidelines for MDR-TB).”

[Please see attached Word document and supplemental material for figure]

Supplemental Figure S1. Changes in World Health Organization treatment guidelines for multidrug-resistant tuberculosis between 1997 and 2022.

Abbreviations: mo: months; XDR-TB: extensively drug resistant tuberculosis; Bdq: bedaquiline; Lzd: linezolid; Am: amikacin; Km: kanamycin; BPaL: bedaquiline, pretomanid, and linezolid; BPaLM: bedaquiline, pretomanid, linezolid and moxifloxacin.

Did the researchers see any impact of the guidance of a 9 month regimen in 2016 on the mean duration of treatment between 2016 - 2019 as that was the first time that WHO recommended a duration of treatment less than 18months?

Response: Regarding mean duration of treatment over time, we did not assess temporal changes as we were attempting to identify patient characteristics for hypothesis generating purposes. Unfortunately, there were few patients included in the study sample (n=223) who were treated in 2016 and later, and it would be difficult to interpret results from such a small sample size when assessing temporal trends. 

For the definition used for extensive disease (line 164) the criteria for CXR matches the WHO definition (Presence of bilateral cavitary disease or extensive parenchymal damage on CXR) but the use of the broad statement of AFB positivity encompasses everything from scanty to 3+ which have different impacts on severity of disease. Did the data sets give the level of smear AFB positivity? If this was given, was scanty/+1 AFB have different durations to 3+ smears? If this is not known, the research team may want to reflect on if this may have had any effect.

Response: The reviewer brings up an important point, unfortunately we do not have the level of smear positivity in our data and only information as to whether the AFB smear was positive or negative. We have added a sentence in the limitations to acknowledge this. 

See the limitation section in the discussion, line 392-394: “We also did not have data on the number of cavities, only the presence or absence, nor did we have data on level of AFB smear positivity (only positive or negative), and were unable to assess what effect this had on duration.”

Line 210 -214 - "Use of capreomycin, kanamycin, moxifloxacin, levofloxacin, PAS, linezolid, clofazimine, Amx-Clv, clarithromycin, or bedaquiline, as well as greater number of drugs, were all associated with longer treatment duration." It is not clear what this sentence is saying - if any of these drugs were individually involved in the regimen then the duration was longer? Most of these drugs are key components of DRTB regimens. This statement also seems to conflict with line 217 "Use of bedaquiline was associated with shorter treatment duration by -0.51 (95% CI -0.87 to -0.15) months".

Response: Thank you for the comment, we have clarified the sentence in the results described below. These results are for the univariable associations, while in the fully adjusted multivariable model bedaquiline was associated with a decrease in treatment duration. 

Please see line 264-267: “Within the treatment regimen of a patient, the use of capreomycin, kanamycin, moxifloxacin, levofloxacin, PAS, linezolid, clofazimine, Amx-Clv, clarithromycin, or bedaquiline, as well as greater number of drugs, were all associated with longer treatment duration in univariable analyses.”

We also modified the text regarding the reversal of effect for the bedaquiline duration estimates in the adjusted analyses.

See lines 287 to 289: “In contrast to univariable regression results, use of bedaquiline was associated with shorter treatment duration by -0.51 (95% CI -0.87 to -0.15) months in adjusted analyses.”

For the statement on line 219 "... and with greater number of drugs used, or use of moxifloxacin, kanamycin, capreomycin, or Amx-Clv." given that prior to the introduction of Bdq in 2013, FLQ and SLI (moxi, Kanamycin and capreomycin) were the backbone of any DRTB regimen so is this saying that the use of these more effective drugs were associated with longer durations?

Response: Thank you for the comment. The result indicates these drugs were associated with longer treatment, when controlling for all other variables in the model. The associations with these drugs should not be interpreted as poorer efficacy, as all patients in the population had successful treatment. Rather, they were just associated with longer treatment duration and as the purpose of this study was hypothesis generation, the reasons for why there was an association with longer treatment should be investigated further. 

In the discussion section the researchers do not mention the latest WHO DRTB guidance (Dec 2022) on the 6 month regimens - BPaLM/BPaL. What does this recommendation for a standard 6 month regimen for BPaLM for adults >14yrs with MDR/RR-TB or with MDR/RR-TB and resistance to fluoroquinolones (pre-XDR-TB), regardless of HIV status mean for the approaches outlined in the article? The BPaLM recommendation does restrict the use of this regimen in those with prior exposure to drugs in the regimen (>1month) so the points made regarding these groups may still be valid but it would be good to see the explicit mention of this dramatic advance in shorter effective treatment for DRTB.

Response: The reviewer brings up an important point regarding the newest regimens that included pretomanid. As our study was not intended to provided evidence to inform clinical recommendations, nor for individual clinical decisions and our results should not be interpreted in that sense. However, for new studies investigating the use of BPaL/BPaLM, our results may be applicable to those studies, in terms of patient populations that should NOT be excluded. We have made note of this in our discussion:

See lines 362-367: “Use of bedaquiline was associated with shorter treatment duration across the majority of subgroups, suggesting that inclusion of patients previously excluded from RCTs9-11,29 or considered ineligible for short MDR treatment in guidelines3 could be included in future trials of shorter bedaquiline and/or pretomanid containing regimens (such as bedaquiline, pretomanid, & linezolid (BPaL), or BPaL plus moxifloxacin, BPaLM) most recently recommended by the WHO3.”

Reviewer #2: The authors have taken a novel approach to try to identify factors associated with treatment duration for MDR/RR-TB. This is a hypothesis generating study. The study is well-written, clearly described with appropriate analyses.

Major comments:

As the data involved in the study are from before 2020, and there is no mention in the article of the 6 month all oral regimens, the article doesn’t really address how the approach might be useful in the future for assessing duration – especially if treatment is given as fixed durations based on regimen.

Response: We appreciate the reviewers point regarding the newest regimens that include pretomanid, which echoes the last point made by Reviewer 1. See our response above. And the revised text lines 362-367 also given above. 

In the supplement 1 under search criteria it states “Studies exclusively in children or of patients treated with short regimens were excluded as these were the topics of two concurrent individual patient data meta-analyses at time of original publication”. If these are indeed excluded it would be helpful to have this stated in the methods. Also then including within the discussion the potential impact or not on the results of this for this methodology.

Response: We thank the reviewer for their response, we have added mention of this in the methods to highlight the exclusion:

See lines 128-129: “Studies exclusively in children were excluded.”

Regarding discussion of populations of children, we have added note of this in our limitations:

See lines 394-396 “Further, studies conducted exclusively in children were excluded from the IPD, and we were not able to assess associations in this population”

Minor comments:

Introduction:

the guidelines for treatment referenced are older guidelines. Suggest reference up to date guidelines from 2022. Line 74-75 states “Current recommended treatment for advanced and extensive MDR-TB is as long as 18-20 months”. This is no longer the case with the 2022 current guidelines. The recommendation that “patients without extensive TB disease and without severe extrapulmonary TB” should not have shorter regimen applies to the 9-month regimens from the 2019 guideline. Suggest rephrase this sentence to clarify how things apply with the current guidelines.

Response: We appreciate the comment. We have updated the reference to the latest WHO guidelines (2022) and updated the wording of the recommendation to reflect the eligibility in the 2022 guideline which recommends the 9-month all oral regimen to “those with no extensive or severe TB disease and no severe extrapulmonary TB” as stated in the guideline.

See line 88-90: “Current recommended treatment from the World Health Organization (WHO) for extensive or severe MDR-TB is as long as 18-20 months3 and entails a high patient burden”

Methods

Line 146 spell out abbreviations at first use: 95% confidence interval (95% CI), and line 163 fluoroquinolone (FQ) and second line injectables (SLI), and 288 ART

Response: Thank you for pointing this out, we have updated the text to include the terms we abbreviated for CI, FQ, SLI, and ART.

Suggest state more clearly whether when referring to MDR-TB you are being inclusive (MDR/RR-TB) or only including those with both isoniazid and rifampicin resistance. From line 95-96 it suggests that also includes rifampicin resistance, however in Methods only ever state MDR-TB which is defined as resistance to both rifampicin and isoniazid. In line 100 it refers to study dataset described in reference 14 – would be easier if state clearly in methods that actually inclusion criteria was rifampicin resistant TB (RR-TB). Would suggest either changing to using RR-TB throughout or MDR/RR-TB throughout and only use MDR-TB when specifically meaning resistance as per definition in in lines 71-72.

Response: We appreciate the response and indeed we did include MDR/RR-TB patients. We have updated the methods and results of the text so we refer to MDR/RR-TB where appropriate throughout.

For the treatment outcomes, 2 references (21 and 22) are mentioned for definitions. These references do have some differences. It would be helpful to the reader to state clearly here whether outcomes given by each site were taken as the outcomes or whether a single set of definitions was retrospectively applied to all date. Also reference 22 is from 2022, while the IPD data was only up til 2019, so not sure if the definitions referred to here were used at all.

Response: We appreciate the comment. We have updated the reference to reflect the WHO definition (WHO 2013) used in each included study that did not use the Laserson et al. definition. In assembling the IPD datasets we verified outcome definitions used by each author/group and harmonized all to the definitions suggested by WHO in 2013, as described in detail in earlier publications (see detailed tables about this in the supplement of Ahmad et al, Lancet). Additionally, we have added clarification on outcome definitions.

See lines 139-143: “From the included studies, we included only patients that had successful (cured or completed) treatment outcomes, as defined elsewhere22,23 and who had their individual treatment duration recorded. We verified outcomes provided by study investigators in their original study, and harmonized these to WHO 2013 definitions,23 as detailed elsewhere (see supplement to Ahmad et al.15).”

Reviewer #3: Winters et al present an analysis of ecological and individual factors associated with successful shorter treatment of patients with MDR TB. Leveraging data from a previously published IPD meta analysis, the authors evaluated data of 6702 patients from 34 different countries. In a site level, the proportion of resistance to FQ-SILI was associated with longer duration, while in at an individual level an adjusted model identified AFB smear, presence of cavities, HIV status and added resistance as factors associated with longer treatment duration. Importantly, the use of bedaquiline was associated with shorter treatment duration. The study is well written and the statistics are appropriate, the topic is important, some comments below:

1. My main concern is that the study relies on data obtained between 2009 and 2018, at that time longer 15-24 month treatment regimens were recommended by WHO. As the authors are aware, current WHO recommendations endorse the use of shorter 6 month all oral regimens using BPaL/BPaLM thus the data presented here albeit interesting may not be applicable to the current MDR TB scenario. The authors should make a point in the discussion into what settings would your data be applicable? Maybe in patients who fail BPaL/BPaLM? In countries without access to pretomanid or bedaquiline? Or in populations were BPaL is not validated yet such as children, pregnancy, end stage renal disease among others.

Response: The reviewer raises an important point regarding the newest regimens that include pretomanid, which echoes the same questions as Reviewer 1 (last point) and Reviewer 2 (first point). See our response to Reviewer 1 (last point) above.

2. Line 77-79 “In the past 10 years, several studies have investigated shorter regimens for treatment “ of MDR-TB in randomized controlled trials (RCTs), but these may not reflect treatment in programmatic settings” This statement is not fully accurate, there is recent data on the applicability of shorter regimens under programmatic conditions (Haley et al pmid: 37249079, Acuna-Villaorduna et al pmid: 37491751). In the US, outcomes of all oral shorter regimens report > 95% treatment success.

Response: We appreciate the reviewer’s comment. Here we were referring to the follow-up and intensity of care that are common in RCTs, compared to programmatic settings. However, we have added mention of the potential success of these regimens when used in programmatic settings and reference to the studies you have mentioned in our discussion.

See lines 427-431: “Although RCTs can provide clearer evidence on optimal duration, these are expensive, time consuming, often lack generalizability, and can test only a limited number of durations and/or regimens at once. As well, evidence from cohorts are useful as potential indicators of success of these regimens in programmatic settings32,33).”

3. The authors included patients with successful treatment outcomes in order to avoid bias which is a reasonable approach, however it could have introduced selection bias. Some of the treatment failure patients who were excluded could have been due to medication side effects and in fact these group may benefit even more of shorter regimens. In fact, in the TB PRACTECAL study, side effects were the main cause (49%) of treatment discontinuation. The authors could consider adding this as a potential limitation

Response: We thank the reviewer for their comment. Indeed, this was a concern of ours in selecting our population of only those who had successful treatment. In our methods we outlined that we assessed the impact this may have on our estimates by analyzing the same model but with inverse probability of selection weights that used a propensity score predicting selection into our study population. We observed no substantial differences between weighted and unweighted estimates (see Results lines 298-301 and Supplemental Table S3). 

However, the effect that adverse events may have on negative treatment outcomes may still have an impact on results, as such we have included mention of this in our discussion. 

See lines 403-418: “Because our population included only those with treatment success, our findings may not be generalizable to all patients with MDR/RR-TB, although we conducted inverse probability of selection weighted analyses which indicated no substantial differences in the excluded vs included populations. However, adverse events may be more common with longer treatment, and contribute to treatment failure (and as such were excluded in this study) yet it is possible that patients with adverse events would benefit even more from shorter treatment.”

4. The authors could expand more on the outcome definition: successful treatment. They refer to the WHO 2022 document (ref 22) who is more an update on shorter regimens and briefly discuss treatment outcomes definition but specifically for all oral shorter regimens which are not the ones the authors evaluate in their analysis. I suggest the authors to add in the text the precise definition of successful outcome they used (cure (negative cultures) without relapse if evaluated at a given time for instance) and how they managed the differences in outcome definitions between different studies in the IPD datasets.

Response: As the reviewer points out, definitions of MDR-TB treatment outcomes have changed over time, particularly with the new short regimens. In assembling the IPD datasets we verified outcome definitions used by each author/group and harmonized all to the definitions suggested by WHO in 2013, as described in detail in earlier publications (see detailed tables about this in the supplement of Ahmad et al, Lancet). 

See lines 139-143: “From the included studies, we included only patients that had successful (cured or completed) treatment outcomes, as defined elsewhere22,23 and who had their individual treatment duration recorded. We verified outcomes provided by study investigators in their original study, and harmonized these to WHO 2013 definitions,23 as detailed elsewhere (see supplement to Ahmad et al.15).”

5. The data on bedaquiline is interesting, I would emphasize more on this. It is not surprising to me that bedaquiline is associated with shorter duration, bedaquiline is highly bactericidal against M. tuberculosis (Yamada et al pmid 36165631) and in animal models can eradicate MTB in 3-4 weeks when used in combination with pretomanid and moxifloxacin. I think the authors could emphasize that their data support the use of bedaquiline for all MDR TB patients.

Response: We thank the reviewer for their comment, and we agree. Bedaquiline is essential in treatment of all MDR-TB patients. We have added emphasis of this in our discussion, but as our study is hypothesis generating we have stated this:

See lines 355-357: “Our finding that treatment duration is shorter when bedaquiline was used, is supported by several studies that have established the efficacy of bedaquiline,3,15,28 which further supports the use of bedaquiline containing regimens for all MDR/RR-TB patients.”

6. SILI abbreviation (line 164) could be defined, I believe it is second line injectables

Response: Thank you for pointing this out, we have added the definition along with the abbreviation for SLI.

6. PLOS authors have the option to publish the peer review history of their article (what does this mean?). If published, this will include your full peer review and any attached files.

Do you want your identity to be public for this peer review? For information about this choice, including consent withdrawal, please see our Privacy Policy.

Reviewer #1: No

Reviewer #2: No

Reviewer #3: Yes: Carlos Acuna-Villaorduna

---

## [Decision Letter · Decision Letter 1]

14 Sep 2023

Identifying patients with multidrug-resistant tuberculosis who may benefit from shorter durations of treatment.

PONE-D-23-17278R1

Dear Dr. Winters,

We’re pleased to inform you that your manuscript has been judged scientifically suitable for publication and will be formally accepted for publication once it meets all outstanding technical requirements.

Kind regards,

Dzintars Gotham

Academic Editor

PLOS ONE

Additional Editor Comments (optional):

Reviewers' comments:

Reviewer's Responses to Questions

**Comments to the Author**

1. If the authors have adequately addressed your comments raised in a previous round of review and you feel that this manuscript is now acceptable for publication, you may indicate that here to bypass the “Comments to the Author” section, enter your conflict of interest statement in the “Confidential to Editor” section, and submit your "Accept" recommendation.

Reviewer #1: All comments have been addressed

Reviewer #3: All comments have been addressed

2. Is the manuscript technically sound, and do the data support the conclusions?

Reviewer #1: (No Response)

Reviewer #3: Yes

3. Has the statistical analysis been performed appropriately and rigorously? 

Reviewer #1: (No Response)

Reviewer #3: Yes

4. Have the authors made all data underlying the findings in their manuscript fully available?

Reviewer #1: (No Response)

Reviewer #3: Yes

5. Is the manuscript presented in an intelligible fashion and written in standard English?

Reviewer #1: (No Response)

Reviewer #3: Yes

6. Review Comments to the Author

Reviewer #1: Thank you for addressing the comments clearly and the addition of the new text and figure highlighting the progression in DRTB care.

Reviewer #3: Thanks for your response. I have no further comments. The authors have replied properly all the queries raised

7. PLOS authors have the option to publish the peer review history of their article (what does this mean?). If published, this will include your full peer review and any attached files.

Reviewer #1: No

Reviewer #3: **Yes: **Carlos Acuna-Villaorduna M.D.

---

## [Editor Report · Acceptance letter]

26 Sep 2023

PONE-D-23-17278R1 

Identifying patients with multidrug-resistant tuberculosis who may benefit from shorter durations of treatment. 

Dear Dr. Winters:

I'm pleased to inform you that your manuscript has been deemed suitable for publication in PLOS ONE. Congratulations! Your manuscript is now with our production department. 

Kind regards, 

on behalf of

Dr. Dzintars Gotham 

Academic Editor

PLOS ONE